# Upper Limits of Downstaging for Hepatocellular Carcinoma in Liver Transplantation

**DOI:** 10.3390/cancers13246337

**Published:** 2021-12-17

**Authors:** Marco Biolato, Tiziano Galasso, Giuseppe Marrone, Luca Miele, Antonio Grieco

**Affiliations:** 1Fondazione Policlinico Universitario Agostino Gemelli IRCCS, 00168 Roma, Italy; marco.biolato@policlinicogemelli.it (M.B.); giuseppe.marrone@policlinicogemelli.it (G.M.); luca.miele@policlinicogemelli.it (L.M.); 2Institute of Internal Medicine, Catholic University of Sacred Hearth, 00168 Rome, Italy; tizianog.1993@gmail.com

**Keywords:** liver cancer, macrovascular invasion, locoregional treatments, transplant, systemic treatments, checkpoint inhibitors

## Abstract

**Simple Summary:**

Currently, most transplant centres worldwide accept patients with hepatocellular carcinoma who underwent successful downstaging. Concurrently, the effectiveness of radiological and systemic therapies used for the downstaging of hepatocellular carcinoma are increasing. It is now more frequently observed that candidates for liver transplantation have an excellent response to downstaging, even if the baseline stage was well beyond the transplantable tumour. Downstaged patients have a higher risk of dropout from the waiting list and post-transplant recurrence if not transplanted in a short time. Since an increasing number of downstaged patients affects the waitlist dynamics, the definition of upper limits of downstaging is becoming a crucial issue. In this narrative review, we summarise current evidence on the downstaging of hepatocellular carcinoma for liver transplantation, including downstaging of patients with macrovascular invasion or extrahepatic metastasis at presentation and employment of the new systemic treatments for hepatocellular carcinoma.

**Abstract:**

In Europe and the United States, approximately 1100 and 1800 liver transplantations, respectively, are performed every year for hepatocellular carcinoma (HCC), compared with an annual incidence of 65,000 and 39,000 new cases, respectively. Because of organ shortages, proper patient selection is crucial, especially for those exceeding the Milan criteria. Downstaging is the reduction of the HCC burden to meet the eligibility criteria for liver transplantation. Many techniques can be used in downstaging, including ablation, chemoembolisation, radioembolisation and systemic treatments, with a reported success rate of 60–70%. In recent years, an increasing number of patient responders to downstaging procedures has been included in the waitlist, generally with a comparable five-year post-transplant survival but with a higher probability of dropout than HCC patients within the Milan criteria. While the Milan criteria are generally accepted as the endpoint of downstaging, the upper limits of tumour burden for downstaging HCC for liver transplantation are controversial. Very challenging situations involve HCC patients with large nodules, macrovascular invasion or even extrahepatic metastasis at baseline who respond to increasingly more effective downstaging procedures and who aspire to be placed on the waitlist for transplantation. This narrative review analyses the most important evidence available on cohorts subjected to “extended” downstaging, including HCC patients over the up-to-seven criteria and over the University of California San Francisco downstaging criteria. We also address surrogate markers of biological aggressiveness, such as alpha-fetoprotein and the response stability to locoregional treatments, which are very useful in selecting responders to downstaging procedures for waitlisting inclusion.

## 1. Introduction

Hepatocellular carcinoma (HCC) is the seventh leading cancer and the second leading cause of cancer death [1], with the incidence of new cases growing worldwide [2]. This growth is due to the increasing number of metabolic and alcohol-related liver diseases.

Liver transplantation (LT) is recommended as the first-line treatment for patients with HCC within the Milan Criteria (MC), for which resection is not possible [3,4]. In Europe and the United States, approximately 1100 and 1800 liver transplantations are performed every year for HCC, out of annual incidences of 65,000 and 39,000 cases, respectively. Mirroring the increasing number of HCC is a continuous growth in the number of LT waiting list registrations. In the United States, that number went from 10.636 in 2015 to 12.677 in 2019, pointing to the resulting organ shortage (srtr.transplant.hrsa.gov accessed on 17 December 2021).

## 2. Transplantation Criteria

The initial experiences of LT for HCC resulted in poor post-transplant survival and high cancer recurrence rates that have been ascribed to suboptimal patient selection [5,6]. In 1996, the introduction of the MC by Mazzaferro et al. [7] ushered in a new era of LT for HCC. Patients who met the following criteria underwent LT: (a) single nodule with diameter ≤ 5 cm; (b) 2–3 nodules, each ≤ 3 cm in size; and (c) no vascular invasion or extrahepatic spread. These patients had a four-year overall survival (OS) rate of 75% and a four-year post-transplant tumour recurrence rate of < 10%. These markers were consistent with those observed for transplant patients with non-cancer indications. Several subsequent studies have confirmed these data within larger patient cohorts [8]. The MC were also rapidly adopted globally and integrated into a prioritisation tool in the United Network of Organ Sharing (UNOS) and the HCC tumour-node-metastasis (TNM) staging system, and Barcelona Clinic Liver Cancer (BCLC) staging systems [9]. Despite this, the MC were considered too restrictive, and several more expansive selection criteria were proposed (Figure 1).

In 2001, F. Yao et al. proposed the University of California, San Francisco, CA, USA (UCSF) criteria [10] of solitary nodule ≤ 6.5 cm, or ≤3 nodules with the largest lesion ≤ 4.5 cm and total tumour diameter ≤ 8 cm, with an OS rate of 75.2% at 5 years and a 5-year recurrence rate of 11.4% after LT. In 2007, a validation of the UCSF criteria was published [8], applying these criteria for LT based on preoperative imaging. In particular, the OS rates at 5 years for patients who met the MC and UCSF criteria were 79% and 64%, respectively, and the tumour recurrence rate was 21.2%. Multivariate analysis revealed that the number of nodules, vascular invasion, and poor tumour differentiation were independent risk factors for reduced post-transplant survival.

In 2009, Mazzaferro et al. [11] published a study based on a collective database that involved 36 centres worldwide and recruited 1556 patients undergoing LT. A population of patients outside the MC was selected, and these patients achieved an OS after 5 years of at least 70%. This is how the “up-to-seven” criteria were established: seven as the sum of the size of the largest tumour (in cm) and the number of nodules without microvascular invasion or extrahepatic spread. More specifically, 283 patients beyond the MC, but within the up-to-seven criteria, had a 5-year OS of 71.2% with a 5-year recurrence rate of 9.1%. The microvascular invasion was an independent predictor of mortality, with the 5-year OS rate dropping to 47.4%. The main limitation of these criteria is that they are based on postoperative data obtained on the explanted liver, focusing on microvascular invasion, making them difficult to evaluate in clinical practice in the absence of surrogate markers of biological aggression.

In subsequent years, several selection criteria were developed that, in addition to the morphological data (number and size of nodules), took into consideration surrogate markers of tumour aggressiveness, such as alpha-fetoprotein (AFP), HCC histological grading, the neutrophil/lymphocyte ratio (NLR), and the behaviour of HCC on PET-CT (Figure 1) [12,13,14,15,16,17,18,19,20,21].

In 2012, Duvoux et al. [13] performed a multicentre study that recruited a training cohort of 537 patients and a validation cohort of 435 patients and proposed the “AFP French Model” based on the number of nodules, the diameter of the largest nodule, and the AFP levels, assigning an overall score between zero and nine. Two groups were selected based on the score: low-risk (score ≤ 2) and high-risk (score > 2) groups. Using this cutoff, the 5-year OS was 67.8%, and the 5-year recurrence rate was 8.8% for the low-risk group. The advantage of this model is that, unlike the UCSF and up-to-seven criteria, it is based on a preoperative evaluation. This study demonstrated that the prediction of tumour recurrence could be significantly improved by incorporating AFP in the selection criteria.

In 2015, Toso et al. [14] proposed another selection criteria incorporating AFP. Patients were transplanted with the following criteria: total tumour volume (TTV) < 115 cm^3^ and AFP < 400 ng/mL. The 4-year OS was similar to that of patients within the MC (78.7% and 74.6%), as was the 4-year post-transplant recurrence rate (4.5% and 9.4%), but these criteria allowed the inclusion of 19.2% more patients beyond the MC. The negative side of this approach was that in the intention-to-treat analysis, the 4-year OS dropped from 71.6% (MC) to 53.8% (TTV/AFP) because of a significantly higher dropout rate in TTV/AFP patients at 12 months after inclusion in the list (18.9% vs. 56.2%).

In 2018, Mazzaferro et al. proposed the Metroticket 2.0 model [21] based on a mathematical model that integrates the MC, UCSF, up-to-seven, and AFP French models, selecting a population with a 5-year OS > 70%. In particular, they showed that a 5-year HCC-related specific survival rate of 72.1% was obtained with the following criteria: if AFP < 200 ng/mL, the sum of the number and size of nodules (in cm) should not exceed seven; if AFP was 200–400 ng/mL, the sum of the number and size of nodules should not exceed five; and if AFP was 400–1000 ng/mL, the sum of the number and size of nodules should not exceed four.

The evolution of imaging techniques led to a refinement and standardisation of HCC diagnostic criteria, summarised in Liver Imaging Reporting and Data System (LI-RADS) protocol released by the American Association of Radiologists, [22]. LI-RADS protocol provides HCC specific diagnostic accuracy for each LI-RADS subclass, ranging from 38% in LR-3 to 74% in LR-4 and 95% in LR-5 nodules. Centonze and coll. observed in a single-centre retrospective study that inclusion of LR-3 and LR-4 (nodules with an intermediate-to-high probability of harbouring HCC), alongside LR-5 and LR-TR-V into the Metroticket 2.0 model, allowed to obtain the best prognostic accuracy [23].

For the criteria in Korea [15], the NCCK has proposed including the behaviour of HCC on PET-CT to the classical number and size of nodules, expressed as total tumour size. In a retrospective study of 280 patients who received living donor grafts, patients with total tumour size < 10 cm and negative PET-CT showed an OS and a post-transplant tumour recurrence rate of 83.6% and 19.3%, respectively, compared with 59.8% and 54.9% for transplant recipients who did not meet these criteria. In patients beyond the MC but within the NCCK criteria, the OS and 5-year post-transplant tumour recurrence rates were 74.6% and 24.5%, respectively, including 10% more patients beyond the MC.

Sapisochin et al. [17] proposed a model based entirely on hepatocarcinoma biopsy grading (G1-G2) and performance status (PS = 0), including any size and number of tumours in the absence of vascular invasion and metastatic lesions. Within these criteria, the 5-year OS and 5-year recurrence rates of patients within the MC were 78% and 13%, respectively, compared to 68% and 30% beyond the MC but within the Toronto criteria. Of the 105 patients beyond the MC, 29 fulfilled the UCSF criteria, and 76 were beyond the UCSF criteria, with a mean number of tumours of 3.5 and a median size of the largest tumour of 3.9 cm.

In 2019 Shimamura et al. reported a retrospective data analysis of the Japanese nationwide survey to expand MC in the context of living-donor LT. The authors proposed the 5-5-500 rule (nodule size ≤ 5 cm in diameter, nodule number ≤ 5, and alfa-fetoprotein value ≤ 500 ng/mL), which was associated with a 5-year recurrence rate of 7.3% and a 19% increase in the number of eligible patients [24].

Finally, in the following years, scores were developed that included different variables with the selection of different risk bands. For example, the TRAIN score proposed by Lai et al. [18] considers the waiting time on the list, the radiological response to locoregional treatment, and the AFP and NLR values. Halazun et al. [19] considered preoperative factors, such as NLR, AFP, and size of the largest nodule, and postoperative factors, such as vascular invasion, grading, and size and the number of nodules on the explanted liver, developing the MORAL score.

Halazun et al. [20] proposed the New York/California (NYCA) score, which takes into account the size of the largest nodule, number of nodules, and the response of AFP to locoregional treatments measured as the difference between the maximum and final pre-LT AFP level.

In 2021, Bhangui et al. [25] published their findings on 405 patients who underwent living-donor liver transplantation using their expanded selection criteria: no extrahepatic disease or major vascular invasion, irrespective of tumour size and number. Preoperative predictors of recurrence were tumour burden beyond UCSF criteria, AFP > 100 mg/mL, and FDG-18 PET avidity. They created a prognostic model stratifying patients into three risk groups: low risk (none of the factors or a single risk factor: recurrence rate 9.3%), moderate risk (two risk factors: recurrence rate 25%), and high risk (three risk factors: recurrence rate 46%).

In 2021, authors from Toronto proposed a machine learning approach to overcome the vast number of predictive factors and the limitation of standard statistical methods. Their proof-of-concept study proposed a ten-variable machine learning algorithm, called the CoxNet model and demonstrated a better prediction of post-LT HCC recurrence risk than the AFP model and MORAL score [26].

## 3. Surrogate Markers of Biological Aggressiveness

### 3.1. Alpha-Fetoprotein

In recent years, markers of HCC biological aggressiveness that predict the main outcomes of LT have been investigated. One of the best-studied is the serum levels of AFP.

The secretion of AFP is associated with the aggressive biological behaviour of the tumour and correlates with vascular invasion and a low degree of differentiation of HCC [13,27,28,29,30]. This correlation is independent and increases with increasing AFP values [13].

High AFP values also correlated with an increased risk of drop-out from the waiting list for LT [17,31] and an increased risk of tumour recurrence and reduced OS after LT [12,16,32,33,34,35]. These outcomes worsen, starting with AFP values above 15 ng/mL [36]. Several cutoffs have been proposed to incorporate AFP into transplantation criteria, but no consensus has been reached on how to combine AFP with the morphological characteristics of HCC. However, it is agreed that AFP values > 1000 ng/mL are significantly associated with worse outcomes after LT [10,29].

In addition to static AFP values, great importance is given to the variation in AFP values over time. In particular, an average monthly increase in AFP greater than 15 ng/mL/month is an independent risk factor for microvascular invasion [27], with an increased risk of post-transplant recurrence and a reduced OS at 5 years [30,37].

An important finding recently emerged that only the latest pretransplant AFP value independently predicted survival [38]. Furthermore, AFP downstaging is associated with good post-LT survival regardless of the maximum AFP level (even if the value was initially > 1000 ng/mL). In fact, according to Halazun et al. [20], patients with AFP greater than 1000 ng/mL who are reported to have levels below 1000 ng/mL with a drop greater than 50% have a disease-free survival at 5 years of 91%, if classified in the low-risk group and 74% if classified in the acceptable-risk group, according to the NYCA score. It has been established, however, that higher AFP values are associated with a higher risk of downstaging failure, estimated as 33.0% in those with AFP ≥1000 ng/mL compared with 15.2% for those with AFP 100–999 ng/mL and 9.3% for those with AFP < 100 ng/mL [39].

Finally, it has been documented that post-transplant AFP levels that do not decrease to < or = 20 ng/mL within two months are indicative of an increased risk of relapse [35].

Des-gamma-carboxy prothrombin (DCP), an abnormal form of prothrombin produced by HCC cells, has been recommended as one of the surveillance biomarkers for HCC in the Japanese guidelines. The use of DCP as a predictor of the risk of recurrence of HCC after living-donor LT was investigated in three retrospective Japanese studies. According to a recent meta-analysis, elevated DCP values were associated with a five-fold increased risk for HCC recurrence after LT [40].

### 3.2. Response to Bridging and Downstaging Treatments

Locoregional treatments are a fundamental tool for managing patients on the waiting list for LT. The latest guidelines of the American Association for the Study of Liver Diseases (AASLD) recommended them as “bridge” therapies in patients on the waiting list for LT to delay the progression of the disease and prevent the consequent “drop out” from the list. The drop-out rate in patients on the waiting list is approximately 7–15% at six months and 25–37% at 12 months [41,42]. It is universally accepted that bridge therapy should be considered when the average waiting time for LT is more than six months [43].

Locoregional treatments have also been indicated to bring back a tumour that has progressed beyond the criteria of transplantation, the so-called “downstaging”. There are still no clear protocols defining the downstaging criteria, universal radiological response criteria, the definition of downstaging response, or post-treatment observational period of stability.

In recent years, the importance of the response to locoregional therapy has been emphasised. Partial or nonresponse to LRT is associated with a higher rate of disease recurrence after LT [44]. The success of LRT depends on the selection of less aggressive tumours, which have better post-LT outcomes in terms of less disease recurrence [45], with results comparable to tumours that already start within the transplantation criteria [28]. In contrast, failure of downstaging often occurs in more biologically aggressive tumours that are more likely to have worse outcomes after LT [46]. Even patients who received LRT and who were not successfully downstaged surprisingly had worse outcomes than patients who not downstaged and did not receive any LRT, despite having significantly fewer advanced stage cancers and less macrovascular invasion [47] In conclusion, downstaging aims to serve as a tool for selecting tumours with favourable biology.

In 2016, Mazzaferro proposed a new stratification of HCC for patients’ candidates to LT. Eight classes of transplantable tumours were identified by combining tumour stage, suitability for LRTs, and tumour response to those treatments. Priority for LT should be linked to the patients’ class [48]. In an Italian validation study, high-risk-classes (including patients with successful downstaging, patients with early recurrence after curative treatment, and patients with the partial response after bridging therapy) were associated with a five-year recurrence risk of 21%, who significantly dropped to <10% if patients were transplanted within two months from restaging [49]. According to the Mazzaferro proposal, restrictive upper limits for downstaging should be established.

### 3.3. Biological Stability

Besides response to downstaging, another fundamental concept is biological stability over time, that is, the stability of the framework reached at the end of downstaging. The observation period after downstaging was three months in many protocols [31], but the optimal length of this observation period for tumour biology is unknown [50,51]. This concept has been filtered into the “ablate and wait” strategy for HCC within or beyond the Milan criteria [52], and subsequently, this concept was supported in the study by Metha et al. [53] and Halazun et al. [54], where (as described below) it was shown that a shorter waitlist time (<3 months) for LT was paradoxically associated with worse post-LT survival.

## 4. Upper Limits for Downstaging?

### 4.1. Introduction

Downstaging patients with hepatocellular carcinoma beyond the Milan criteria is increasingly being used to expand access to life-saving transplants.

The goal of liver transplantation is to provide liver recipients with the maximum possible benefit from the limited resources of organs of deceased and living donors in a fair, ethical, and convenient way. As mentioned above, requests for the organ are increasing from year to year, and not all requests can be satisfied. The advantage of transplantation over liver resection is undoubted, especially in the intermediate stage (BCLC B) in the absence of vascular invasion [55]. In the context of organ shortages for liver transplantation, an extension of the boundaries of transplantation for HCC must consider the benefit for individual HCC patients and the consequences for all potential liver recipients. This led to the recommendation of the international consensus conference [56] in which the experts are inclined to accept the expansion criteria beyond the MC provided that the 5-year survival after transplantation was comparable to non-HCC patients and to accept LT after successful downstaging if the 5-year survival was comparable to that of HCC patients who met the criteria for liver transplantation without requiring downstaging. Furthermore, patients with a worse prognosis outside the Milan criteria may be considered for liver transplantation if the dynamics on the waiting list allow it without undue prejudice to other recipients with a better prognosis. EASL guidelines also affirmed that patients beyond the Milan criteria could be considered for LT after successful downstaging to within the Milan criteria and within defined protocols.

The price to pay to reach the balance between the increasing number of patients on the waiting list with the expansion criteria and the achievement of comparable outcomes between patients initially within and beyond the criteria is the increased risk of drop-out from the waiting list [14,31,57].

### 4.2. Downstaging Protocols

A systematic review evaluating the outcomes of 13 downstaging studies showed a pooled post-LT HCC recurrence rate of 16%; heterogeneity in survival reporting limited the pooling of the data, but generally, the 4-year to 5-year survival was 70% to 90% [58]. Widening the selection criteria is essential to increase the number of eligible patients to meet the increased demand for organs. At the same time, these politics may be accompanied by an increased rate of waitlist dropout in the expanded group [14] and worse outcomes after LT. The optimal selection of patients for downstaging protocols includes an assessment of tumour burden, liver function, and surrogate markers for tumour biology, including alpha-fetoprotein. To date, no standardised criteria for downstaging have been defined, and most of the present studies have adopted heterogeneous criteria, making them difficult to compare. In particular, we have to clearly define the entry criteria, such as size and number or the total tumour volume of HCC, biological, pathological, and molecular markers, endpoint criteria of successful downstaging such as radiological (degree of necrosis, decrease in size) and biological (AFP) endpoints, and the definite time between downstaging and listing for transplant (Table 1 and Table 2).

The first authors who proposed a downstaging protocol with a prospective study were Ravaioli et al. [28] in 2008. In detail, they proposed the following downstaging criteria for patients beyond the Milan criteria: single HCC 5–6 cm, two HCCs ≤ 5 cm, or less than six HCCs ≤ 4 cm and the sum of diameter ≤ 12 cm, but within the Milan criteria with a stability period of three months, including AFP stably below 400 ng/mL after the downstaging procedures. The outcome of downstaged patients beyond the MC (*n* = 48) was compared with patients within the MC (*n* = 129) after LT and the first evaluation according to an intention-to-treat analysis. There was no significant difference between the two groups, with ITT survival rates at the end of the study of 56.3% and 62.8%, respectively. Downstaging was achieved in 89.6% of cases, with a drop-out rate of 33%. The mean wait time at LT was 143 days for the MC group and 185 days for the downstaged MC group. Furthermore, the only variable independently related to tumour recurrence was an AFP level higher than 30 ng/dL. When the AFP was < 30 ng/mL, surprisingly, microvascular invasion and poor differentiation were not associated with an increased rate of recurrence. They observed an increased progression rate in the downstaged MC group (27.1%) compared with the MC group (11.6%), despite a nonsignificant difference in waiting time on the list and in terms of recurrence-free survival and overall survival after LT. This increased risk of progression translates into prioritising patients after successful downstaging.

In 2008, a second prospective study was published by Yao et al. [59] proposing the UCSF-DS protocol for patients with tumour burden beyond the MC but within the following criteria: one lesion >5 cm and up to 8 cm; two to three lesions with at least one lesion >3 cm and not exceeding 5 cm, with a total tumour diameter up to 8 cm; or four to five lesions with none > 3 cm, with a total tumour diameter up to 8 cm. A minimum observation period of three months after downstaging was required before LT. A sample of 61 patients was recruited in this prospective study. Tumour downstaging was successful in 43 patients (70.5%). The Kaplan–Meier ITT survival at one and four years after downstaging were 87.5% and 69.3%, respectively. The 1-year and 4-year post-transplantation survival rates were 96.2% and 92.1%, respectively, in the absence of HCC recurrence.

These data were subsequently confirmed by a study with a larger sample and with a longer post-LT follow-up [31]. Notably, a group of patients always within the MC, T2 (*n* = 488), was compared with a group of patients beyond the MC (*n* = 118) and downstaged according to the UCSF-DS protocol. The authors demonstrated that successful downstaging of HCC to within T2 criteria was associated with a low rate of HCC recurrence and excellent post-transplant survival, comparable to those meeting T2 criteria without downstaging with 5-year ITT survival rates of 56.1% and 63.3%, respectively (*p* = 0.24). Tumour downstaging was successful in 65.3% of patients. The drop-out rate was 38.1%, with a mean waiting list time for LT after a downstaging of 9.8 months. The cumulative incidence of dropout in the downstaging group was significantly higher than that in the T2 group, likely related to a greater initial tumour burden. Factors predicting dropout in the downstaging group included pretreatment AFP > 1000 ng/mL.

In a single-centre retrospective study, Lee et al. [46] analysed a sample of 213 HCC T3 patients downstaged into the MC with a success rate of 67.6%. The dropout rate was not reported. The 5-year OS of patients with successful downstaging was significantly higher than that of patients with downstaging failure and patients with initial T1 or T2 stage and progression, probably because of the selection of the most biologically favourable tumours. Significant independent predictors (at pathology assessment) for successful downstaging in the multivariate analysis were the last AFP before LT < 70 ng/mL (OR: 2.18), the absence of macrovascular invasion (OR: 4.82), and the absence of satellite nodules (OR: 6.82).

Mehta et al. [39] conducted the first multicentre study using the UCSF protocol in United Network for Organ Sharing (UNOS) Region 5. A total of 187 patients were successfully downstaged, with an 83.4% success rate. During the waitlist, 36.4% of patients were lost to follow-up. The 5-year intention-to-treat survival was 55.4%. The mean waiting list time was not specified. In the logistic regression analysis, the only factors predicting the inability to ever achieve tumour downstaging were pretreatment AFP > 1000 ng/mL and Child–Pugh B/C. In particular, treatment failure was observed in all Child–Pugh B/C patients with pretreatment AFP ≥ 1000 ng/mL.

Primarily based on these results, a new UNOS policy in early 2017 adopted the UCSF/Region 5 inclusion criteria for downstaging (UNOS-DS) [31]. Patients with an initial tumour burden meeting the UNOS-DS inclusion criteria who achieve successful downstaging within the MC are eligible to receive automatic approval for the Model for End-Stage Liver Disease (MELD) exception point, a prioritisation for LT. Given that Region 5 (southwest states) is one of the U.S. regions with the longest waiting time for LT, and given that waiting times have been linked to different post-LT outcomes in HCC, [53] the Region 5 results may not be generalisable to other regions.

Several studies have not placed upper limits on the number and size of HCC nodules for downstaging. The only limits were the absence of vascular invasion and extrahepatic spread [60,61,62,63,64,65,66,67]. A flaw of these studies is that the outcome analysis was not performed with the intention-to-treat purpose in most cases. Furthermore, they were very heterogeneous in the downstaging protocols. The data are therefore not directly comparable; however, they are helpful in opening a debate.

Graziadei [60], in a prospective study on a sample of 15 patients beyond the MC, had a downstaging rate of 73.3% and a 5-year ITT survival rate of only 31%, mainly because of a 5-year recurrence rate of 30%. In comparison, the 48 patients within the MC had a 5-year ITT survival rate of 94%, with a 5-year recurrence rate of 2.4%.

Cillo [61], in his prospective single-centre study, did not place limits on the size or number of tumours in the downstaging protocol but excluded tumours with poor histological differentiation. The intention-to-treat survival rates at 3 years were 85% in Milan-out patients and 69% in Milan-in patients. None of the 68 transplanted patients had recurrent HCC after a median 16-month follow-up. These data were surprising but attributable to the fact that there was greater sensitivity in giving priority to progressing tumours, as evidenced by the lower average wait for transplantation in the Milan-out group (10.2 months) than in the Milan-in group (12.4 months).

Less promising in terms of outcomes are the studies by De Luna [64], Jang [65], and Barakat [66], which showed downstaging success rates of 63%, 41.5% and 56%, respectively, with high drop-out rates and poor ITT survival. Barakat offers an interesting insight into the clear difference in 2-year ITT survival between successfully downstaged patients and those who are not downstaged (78% vs. 7%). These data underline the fact that successful downstaging depends on tumours with more favourable biology.

Most of the published studies on downstaging have used the MC as a successful endpoint of downstaging. Recently, a retrospective multicentre study in China [67] investigated the endpoint of downstaging the Hangzhou Criteria (HC): (a) reduction in total tumour diameter ≤ 8 cm or (b) total tumour diameter > 8 cm with grade I or II tumour differentiation but AFP level ≤ 400 ng/mL. Successfully downstaged patients had OS and recurrence rates after transplantation comparable to those always within the HC, while the OS rates of patients downstaged to within the MC and within the HC were the same.

Little is known about the results of downstaging for patients with initial tumour size and number exceeding the UCSF-DS criteria. Sinha et al. [57], in the study “All-Comers Protocol”, aimed to evaluate ITT survival and post-LT outcomes for downstaging in patients with an initial tumour burden beyond the UCSF-DS criteria, defined as “all-comers” (AC), and compared these outcomes with those of patients meeting the UCSF-DS criteria. This study included 74 patients enrolled in the AC protocol for downstaging and 133 patients enrolled in the UCFS-DS protocol. Patients with vascular invasion or extrahepatic spread were excluded from the AC protocol. The criteria for successful downstaging were the same as those of the UCSF-DS but differed in the stability waiting time after downstaging, which was six months (instead of three months), and in a new exclusion criterion after downstaging in addition to disease progression, which was the appearance of new HCC nodules. Successful downstaging to Milan was lower in the AC group (64.8%) than in the UCSF-DS group (84.2%). The cumulative probability of drop-out was higher for the AC group than for the UCSF-DS group, both at 1 year (53.5% versus 25.0%) and 3 years (80.0% versus 36.1%). Factors predicting drop-out included the sum of tumour number and largest tumour diameter greater than 8 (HR 1.79, *p* = 0.049) and Child–Pugh class B and C (HR 2.54, *p* = 0.001). At the end of the study, only 13.5% of patients in the AC group received LT versus 59.0% in the UCSF-DS group due to the higher drop-out rate. ITT survival was lower for the AC group than for the UCSF-DS group at 1 year (77.4% versus 85.5%) and 5 years (21.1% versus 56.0%). It is possible, however, that the stricter criteria of the AC protocol affected the study results.

Mehta et al. [53] recently applied the principles of the All-Comers study to patients in the UNOS database, including 3819 patients. In addition to identifying the three groups (MC, UCSF-DS, and AC-DS), the eleven participating UNOS regions were divided based on the median time from initial listing to LT in long-wait regions (LWR, >9 months), mid-wait regions (MWR, 3–9 months), and short-wait regions (SWR, <3 months). The authors observed similar 3-year post-LT survival between the Milan and UNOS-DS groups but significantly worse survival in the AC-DS group. These results supported the need for restricting the upper limits of tumour burden for downstaging. Furthermore, they found that a shorter waitlist time (SWR group) and AFP ≥100 ng/mL at LT were associated with worse post-LT survival in the downstaged groups (both UCSF-DS and AC-DS). The same authors in another study [68] analysed the risk of waitlist dropout among patients with HCC in long-wait regions (LWRs) and created a drop-out risk score based on four variables: numbers of nodules, AFP > 20 ng/mL, increasing Child–Pugh grade and model for end-stage liver disease-sodium scores. This risk score was able to stratify the 1-year cumulative incidence of drop-out from 7.1% with a score of <−7 to 39.5% with a score of >23. Patients with a dropout risk score of >30 had a 5-year post-LT survival of 60.1% compared with 71.8% for those with a score of <−30.

In 2020, Mazzaferro et al. published the multicentre, randomised, controlled XXL trial on downstaging in Lancet Oncology [69]. The upper limit for downstaging eligibility was determined based on the prediction of outcome (at least 50% survival at five years on the Metroticket Calculator [11]) and not based on predetermined cutoffs for size and number of nodules at presentation, in the absence of vascular invasion and extrahepatic spread. AFP values before transplantation were taken into consideration by identifying the threshold value of 400 ng/mL after downstaging, above which liver transplantation was not allowed. The response to downstaging was evaluated using the RECIST radiological criteria with an observation period of three months before enlisting. It was considered a successful procedure when a partial or complete response to the LRT was achieved. Seventy-four patients outside the MC were recruited and downstaged. The downstaging success rate was 73%, with a complete response in 43% of patients and a drop-out rate of 16.6%. The remaining 45 patients were randomised and assigned into two groups: the control group and the LT group. At follow-up, a statistically significant reduction in ITT survival was observed in the control arm. The 5-year OS was 77.5% in the LT group and 31.2% in the control group. Unfortunately, the study was closed early because of the national revision of transplantation priorities for HCC.

Recently, Metha et al. [70] published the results of the multicentre downstaging UNOS protocol, including 209 patients with HCC evaluated from 2016 to 2019. The probability of successful downstaging to the Milan criteria was 87.7%, while the dropout rate at 2 years from the initial downstaging procedure was 37.3%. Both chemoembolisation and yttrium-90 radioembolisation were equally effective in terms of radiological response, probability of or the time to successful downstaging, waiting list dropout, or LT. In the explant, 42.8% of patients exceeded the Milan criteria (understaging), and 17.5% had a vascular invasion.

### 4.3. Successful Downstaging in Patients with Macrovascular Invasion at Presentation

The macrovascular invasion has historically been considered a contraindication for radical surgical procedures, such as liver transplantation for hepatocellular carcinoma (HCC), due to the 20-fold greater risk of recurrence and 4.9-fold worse OS [12]. In other studies, this gap was smaller but still significant, probably due to the large sample size [16], with a 2.3-fold worse OS and 1.8-fold higher post-transplant recurrence rate. Evidence of macrovascular invasion in the explanted liver was associated with a 1-year OS of 33% [10] and was an independent risk factor for unsuccessful downstaging [46]. Macrovascular invasion is considered a manifestation of an advanced-stage tumour and therefore has remained a contraindication in current guidelines [3].

In 2008, Chapman [62] published a single-centre preliminary study on downstaging comparing two groups of patients (within and beyond the MC) and including some patients with tumour thrombosis of the portal branches. The downstaging rate was low, at approximately 22%, but successful transplant recipients showed a 5-year OS of 93%. In 2017, these data were confirmed in another study [63] with the same downstaging protocol, including patients with portal branch vascular invasion. In accordance with the previous data, the downstaging success rate for MC was 30%. Downstaged patients showed 5-year OS and recurrence rates similar to those of Milan-in patients.

More recently, some groups from Korea [71] and India [72] have retrospectively reviewed single-centre results of living donor liver transplants in patients with HCC and portal vein tumour thrombosis, with and without downstaging procedures with stereotactic body radiotherapy and tumour ablation. Their results suggest that a subset with segmental PVT (not in the portal vein trunk) could achieve a 5-year OS of approximately 50%.

In 2020, Assalino et al. [73] published a multicentre retrospective study that explored the outcome of 30 HCC patients transplanted after a complete radiological regression of vascular invasion by locoregional or surgical therapies. The 5-year OS was 60%, and the recurrence rate was 11% in the subgroup of patients with pretransplant AFP < 10 ng/mL.

Last year, the group in Bologna [74] reported results from the “superdownstaging” protocol, which included HCC patients with segmental portal vein tumour thrombosis treated with yttrium-90 TARE and enlisted for deceased donor LT in case of complete and sustained (six months) radiological response. The 5-year OS was 60%, but recurrence was a concern.

Differentiating tumour versus bland portal vein thrombosis on imaging can be challenging, even for expert radiologists. Fine-needle biopsy of thrombosis is the gold standard, but it is often contraindicated in the setting of coagulopathy and ascites. In 2019, authors from UCSF proposed A-VENA criteria for noninvasive differentiation of tumour versus nontumour thrombosis in the setting of liver transplant candidates with HCC. The presence of at least three out of five criteria (AFP > 1000 ng/dL, Venous expansion, thrombus Enhancement, Neovascularity, and Adjacent to HCC) had 100% sensitivity and 94% specificity in the diagnosis of tumour portal vein thrombosis [75]. Another approach is Contrast-enhanced ultrasound, which can detect neovascularity of tumour thrombosis with an excellent safety profile. A meta-analysis demonstrated 94% sensitivity and 99% specificity in the diagnosis of tumour portal vein thrombosis [76].

Taken as a whole, these data seem to question the validity of vascular macroinvasion as a definitive contraindication to liver transplantation. However, we need further research both in the correct selection of patients (pretransplant AFP cutoff, extent of macroinvasion vascular) and in the evaluation of the duration of the radiological response (perhaps six months to one year) to reach more stringent and reliable inclusion criteria [77].

### 4.4. Successful Downstaging in Patients with Extrahepatic Metastasis at Presentation

Similarly, the presence of extrahepatic metastases has always been a contraindication to liver transplantation. The existing guidelines only recommend targeted therapy, systemic chemotherapy, or the best supportive care for HCC patients with extrahepatic metastasis [3]. The role of resection of metastatic tumours in HCC patients with extrahepatic metastasis remains unclear. Two recent reviews [78,79] showed that the majority of experts believe that resection of lung metastases has a survival benefit when the hepatic lesion is resected or controlled [80], and in selected cases, the reported long-term survival may be achieved by resecting metastasis at sites of the abdominal lymph node [81,82,83], adrenal gland [84], and peritoneum [85]. The encouraging results demonstrate that highly selected patients may be suitable candidates for these radical curative pursuits. Therefore, are these patients also candidates for liver transplantation? Interestingly, Tae-Yong Ha et al. [86] reported patients treated with adrenalectomy for metachronous adrenal metastases after surgical resection or LT for HCC. It was not a randomised trial; however, in these selected cases, the OS and recurrence risk were 20% at 5 years and 100% at 18 months, respectively, in the group previously undergoing liver resection and 85% and 28% at 5 years, respectively, in the group previously treated with LT. Therefore, surgical treatment combined with personalised systemic treatment may show a survival benefit for selected patients with extrahepatic spread whose primary tumour has been adequately controlled. To date, no trial or case report in the literature shows data on liver transplantation after the downstaging of metastatic HCC.

### 4.5. Downstaging HCC Patients to within Transplant Criteria after Systemic Treatments

BCLC classification defines advanced-stage HCC patients as those with macrovascular invasion, extrahepatic metastases, or symptoms associated with cancer. These patients have a very poor prognosis and are generally excluded from a transplant program due to the high risk of post-transplant recurrence, which affects their post-transplant survival. For these patients, current guidelines recommend systemic therapies.

In 2008, sorafenib, an oral multityrosine kinase inhibitor, was the first drug to demonstrate a survival benefit in patients with advanced HCC [87] and was approved for first-line therapy of HCC. In the SHARP trial, the median overall survival was 10.7 months in the sorafenib group and 7.9 months in the placebo group. Sorafenib was also used with patients in the intermediate stage who failed to respond to locoregional treatments. A prospective multicentre study in Italy demonstrated that the overall survival was 8.4 months in BCLC-C patients and 20.6 months in BCLC-B patients, with 8% of patients achieving complete or partial radiological response [88].

In 2018, lenvatinib was the second oral multikinase inhibitor approved as first-line therapy for HCC after demonstrating noninferiority in terms of the median overall survival (13.6 months for lenvatinib vs. 12.3 months for sorafenib) and a higher objective response rate (ORR) than sorafenib (24% vs. 9%) [89].

The combination of an anti-PD-L1 antibody (atezolizumab) and an anti-VEGF antibody (bevacizumab) was tested as first-line therapy versus sorafenib in a global, open-label, phase 3 trial [90] and showed better overall survival at 12 months (67.2% vs. 54.6%) and progression-free survival (6.8 months vs. 4.3 months) than sorafenib.

Other oral multikinase inhibitors, including regorafenib and cabozantinib, were approved as second-line therapies for HCC after first-line sorafenib treatment. In 2017, the results of the RESORCE trial were published, showing a survival benefit in the regorafenib arm (10.6 months) versus placebo (7.8 months) for patients who progressed after sorafenib [91]. In 2018, again in a second-line setting, cabozantinib showed a median overall survival of 10.2 months versus eight months for patients receiving placebo [92]. Another second-line option is ramucirumab, a monoclonal antibody that antagonises VEGF receptor 2. Ramucirumab was tested in a phase III trial in patients in sorafenib-progressed patients with AFP > 400 ng/mL, showing a median overall survival of 8.5 months compared with 7.3 months for patients receiving placebo [93].

After successfully treating numerous cancers, checkpoint inhibitors have also recently been studied for HCC.

Nivolumab, a programmed cell death protein-1 (PD-1) immune checkpoint inhibitor, was studied as second-line therapy for HCC in a phase one-half dose escalation and expansion trial [94]. It showed a median overall survival for patients in the dose-escalation phase of 15 months and an ORR of 14%. Preliminary results from CheckMate 040 [95] showed a promising ORR of 31% for second-line nivolumab plus ipilimumab combination therapy. Preliminary results showed comparable survival rates between nivolumab and sorafenib or lenvatinib in the first-line setting [96]. In phase 3 Checkmate-459 trial, treatment with nivolumab resulted in a 4% complete response rate and 12% partial response rate [96].

Pembrolizumab, another PD-1 inhibitor, showed an ORR of 17% in a phase II trial [97]. In phase 3 KEYNOTE-240 trial, treatment with pembrolizumab resulted in a 2.2% complete response rate and 16.2% partial response rate [98].

Despite the emergence of systemic therapies, their role remains under study in tumour downstaging until transplantation. In all the trials presented, the response rate to cancer under systemic therapy was low, and a complete response was observed only in rare cases. However, when the patient meets the transplant criteria, the question is whether it is appropriate to put the patient on a transplant waitlist. What must be the minimum duration of the radiological response be before considering whether the patient is eligible for transplant? Is it ethical not to transplant this patient with respect to the other patients on the waitlist [99,100]?

In the literature, the only data available on patients successfully downstaged after systemic treatments came from case reports and case series. For example, Vagefi et al. [101] described an HCC patient with portal vein tumour thrombosis and an AFP value of 194,000 ng/mL. This patient was treated with sorafenib and presented a complete radiological response with normalisation of AFP levels. The patient was subsequently placed on a waiting list for liver transplantation, but it was not declared whether the patient had been transplanted. Kermiche–Rahali et al. [102] described another patient with large HCC in the left liver associated with portal vein tumour thrombosis. After nine months of sorafenib treatment, reassessment showed that the tumours had decreased in size with recanalisation of the portal vein. A left lateral hepatectomy was performed, and pathology showed complete tumour necrosis. Yoo et al. [103] described a patient with large right lobe HCC and single lymph node metastasis who was downstaged to the UCSF criteria after radiotherapy, chemotherapy and sorafenib treatment and transplanted from a living donor without recurrence for 11 months. Jeng et al. [104] described a patient with HCC and right main portal vein tumour thrombosis with a complete radiological response after percutaneous alcohol injection, radiofrequency ablation, eight sessions of transcatheter hepatic arterial chemoembolisation, radiotherapy, and sorafenib. He was transplanted from a deceased donor and showed no tumour recurrence for 20 months. Recently, Huang et coll. [105] published a case series including 13 advanced HCC patients with portal vein tumour thrombosis receiving the combination therapy of sorafenib, camrelizumab, transcatheter arterial chemoembolisation and stereotactic body radiation therapy, showing macrovascular invasion regression in 33% of cases. Responder patients underwent hepatectomy.

A Chinese retrospective study described 60 patients with advanced HCC (55% with macrovascular invasion, 27% with extrahepatic disease) treated with lenvatinib combined with anti-PD-1 antibodies, showing an ORR of 33%. The response rate in patients with portal vein tumour thrombosis was greater than the response rate in patients with intrahepatic lesions [106]. In 2021, Yang et al. [107] described a series of nine patients with advanced HCC and oligometastasis (less than five metastases) who were successfully downstaged after treatment with lenvatinib plus anti-PD-1 antibody and transarterial chemoembolisation. These patients underwent resection of residual hepatocellular tumours, and the recurrence rate was 22% at 10 months.

Checkpoint inhibitors target coinhibitory pathways and promote T-cell activity, thus inducing immune control of cancer. Since programmed cell death protein 1 (PD-1) is abundantly expressed on human liver allograft-infiltrating T-cells, there is concern about using checkpoint inhibitors in liver transplant patients because T-cells can proliferate considerably after programmed death-ligand 1 blockade [108]. Experience with checkpoint inhibitors in solid organ transplant recipients is limited, and there is a risk of graft rejection. Gassmann reviewed the limited literature available, including some patients treated for HCC recurrence after liver transplantation and calculated risk for graft loss ranging from 36% to 54% [109]. Tabrizian et al.. [110] described nine patients with HCC who were listed and successfully transplanted at Mount Sinai Medical Center after receiving nivolumab as an element of pretransplant tumour treatment and observed that only one patient developed mild acute rejection. The authors speculated that the substantial transfusion requirement during transplantation might result in the acute clearance of a proportion of serum nivolumab, accelerating the elimination of the drug and its effects. Schwacha–Eipper et al. [111] described a patient with intermediate HCC who progressed after sorafenib and regorafenib treatment and achieved partial response to nivolumab to meet the Milan criteria. The patient was transplanted without recurrence or rejection at the one-year follow-up. The authors, considering the four-week half-life of nivolumab, decided to stop treatment six weeks prior to activation on the waiting list to avoid rejection.

Given the paucity of data, further prospective studies are needed to investigate the efficacy and safety of checkpoint inhibitors in HCC patients treated with the purpose of downstaging for liver transplantation. Given the potential risk of graft rejection, it is necessary to study protocols regarding the minimum time of therapy discontinuation before the patient’s activation on the list for transplantation is considered sufficiently safe.

## 5. Perspectives

While there is a general consensus on exceeding the MC for selecting patients for liver transplantation, most transplant centres rely on extended criteria well established in the literature, such as the up-to-seven criteria, and adopt consolidated downstaging protocols, such as the UCSF-DS criteria. However, there is growing pressure to consider listing patients who were initially excluded due to a high disease burden but who subsequently had good responses to the locoregional or systemic treatments currently available that make them fall within the transplant criteria. We mention examples such as HCC patients with large nodules who respond to radioembolisation or patients with macrovascular invasion or even extrahepatic metastasis who respond to new systemic treatments with a complete and sustained radiological response for portal thrombosis or even extrahepatic metastasis.

A combination of systemic treatment with locoregional therapy would be one of the effective strategies for the downstaging of HCC. The possibility of combining different treatments offers various opportunities; for example, it is possible to concentrate locoregional treatments on the largest nodules, with the aim of downstaging the patient within the transplantation criteria, and to treat the other nodules by systemic therapy in order to save time on the waitlist and to reduce the risk of dropout. Clearly, such an approach is conditioned by the patient’s excellent hepatic functional reserve.

To set a framework for these challenging scenarios, it would be crucial to know the 5-year post-transplant survival rate and the relapse rate, but as we have seen in this review, this information is not currently available, except for patients with macrovascular invasion. The study of Assalino et al. [73], however, needs prospective multicentre confirmation, with an analysis of the potential impact of the drop-out risk for the other patients on the waitlist. As we learned from the All-Comers study [57], the list dynamics (especially if the waiting time is long) are of great importance in determining the success or failure of a downstaging program.

In accepting the transplant candidacy of downstaging responsive patients, we must rely on the lessons learned about biological aggression markers and try to fit them into the local waiting list dynamics. For example, as suggested by the study by Mehta et al. [53], in long waiting lists, time can be used to one’s advantage to exclude those patients who experience radiological disease progression, according to the criterion of loss of biological stability. This criterion cannot be applied in local realities with a short waiting list, and in the latter, it becomes essential to emphasise the changes in AFP, even if there is still no uniformity in the cut-off value or in the decision-making trend to be used.

## 6. Conclusions

We eagerly await more data, preferably from prospective studies on transplanted patients after successful downstaging of advanced HCC with macrovascular invasion or extrahepatic metastasis at presentation. However, until then, we believe it is fairer and more responsible not to place these patients on transplant waiting lists outside of clinical trials.

## Figures and Tables

**Figure 1 cancers-13-06337-f001:**
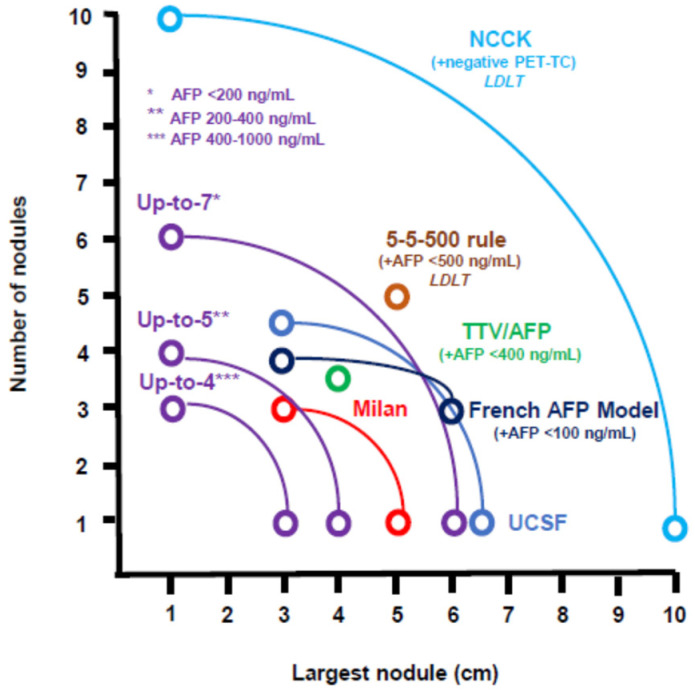
Different expansion of criteria for liver transplantation for Hepatocellular carcinoma. Abbreviation: AFP = Alpha-fetoprotein, LDLT = Living Donor Liver Transplantation). NCCK = National Cancer Center Korea; TTV = Total Tumour Volume; UCSF = University of California, San Francisco, CA, USA.

**Table 1 cancers-13-06337-t001:** Protocol characteristics of studies on downstaging of hepatocellular carcinoma before liver transplantation.

*Authors* *(Year)*	*Sample*	*Study Design*	*Downstaging Criteria*	*Downstaging* *Success Rate*
*Studies with upper limits of downstaging by size and number of tumours*				
Yao (2008)	61	prospective monocentric	One tumour 5–8 cm or 2–3 tumours 3–5 cm with TTD up to 8 cm or 4–5 tumours < 3 cm, with TTD up to 8 cm	70.50%
Ravaioli (2008)	48	prospective monocentric	One tumour 5–6 cm or Two tumours ≤ 5 cm or <6 tumours ≤ 4 cm and TTD up to 12 cm	89.60%
Lei (2013)	58	retrospective monocentric	One tumour ≤ 8 cm or 2–3 tumours ≤ 5 cm and a TTD up to 8 cm	/
Yao (2015)	118	retrospective monocentric	One tumour 5–8 cm or 2–3 tumours 3–5 cm with TTD up to 8 cm or 4–5 tumours < 3 cm, with TTD * up to 8 cm	65.30%
Metha (2018)	187	retrospective multicentric	UNOS protocol: One tumour ≤ 8 cm or 2–3 tumours ≤ 5 cm or 4–5 tumours ≤ 3 cm with TTD ≤ 8 cm	83.40%
Sinha (2019)	207	retrospective multicentric	UCSF-DS vs. AC-DS	84.2% vs. 64.8%
Metha (2020)	543	retrospective multicentric	UNOS-DS vs. AC-DS	/
Lewandowski (2009)	86	retrospective monocentric	OPTN T3	TACE 31% TARE 58%
Lee (2020)	247 (LDLT)	retrospective monocentric	OPTN T3	68%
*Studies with no upper limits of downstaging by size and number of tumours but no vascular invasion or extrahepatic spread*				
Mazzaferro (2020)	23 LT vs. 22 control	multicentric randomised trial	5-y estimated post-LT survival > 50%, (Child–Pugh A-B7) AFP < 400 ng/mL	73%
Cillo (2007)	40	prospective monocentric	No G3. No upper limits *	/
Graziadei (2003)	15	prospective monocentric	No upper limits *	73.30%
Otto (2006)	62	prospective monocentric	No upper limits *	54.80%
De Luna (2009)	27	retrospective monocentric	No upper limits *	63%
Jang (2010)	386	retrospective monocentric	No upper limits *	41.50%
Barakat (2010)	32	retrospective monocentric	No upper limits *	56%
Green (2013)	22	retrospective monocentric	No upper limits *	77%
Toso (2015)	39	retrospective multicentric	No upper limits *	/
Hangzhou (2020)	206	retrospective multicentric	No upper limits * Hangzhou Criteria °	39.50%
Kardashian (2020)	465	retrospective multicentric	No upper limits *	/
*Studies with no upper limits of downstaging for vascular invasion*				
Chapman (2008)	136	prospective monocentric	No upper limits, including tumour thrombosis of the portal vein branch #	22.30%
Chapman (2017)	63	retrospective monocentric	One tumour > 5, 2–3 tumours > 3, or > 4 lesions with any size, any tumour stage plus intrahepatic portal or hepatic vein involvement #	42.00%
Assalino (2020)	30	retrospective multicentric	No upper limits, including the presence of macrovascular invasion #	/

AC: All-Comers, AFP = Alpha-fetoprotein, DS = Downstaging, LDLT = Living Donor Liver Transplant, LT = Liver Transplant, OPTN = Organ Procurement and Transplantation Network, TACE = Transarterial chemoembolization, TARE = Transarterial Radioembolization, TTD = Total Tumour Diameter, UCSF = University of California San Francisco, San Francisco, CA, USA, and UNOS = United Network for Organ Sharing. * for size and number of tumours. No vascular invasion or extrahepatic spread. ° Hangzhou Criteria, Hangzhou, China: (a) reduction in total tumour diameter ≤ 8 cm or (b) total tumour diameter > 8 cm with grade I or II tumour differentiation but AFP level ≤ 400 ng/mL. # No extrahepatic spread.

**Table 2 cancers-13-06337-t002:** The outcome of studies on the downstaging of hepatocellular carcinoma before liver transplantation.

*Authors* *(Year)*	*Dropout Rate*	*Overall* *Survival*	*Recurrence Free Survival or* *Recurrence Rate*	*Intention to Treat Survival*
*Studies with upper limits of downstaging by size and number of tumours*				
Yao (2008)	/	4-y 92.1%	4-y RFS 100%	4-y 69.3%
Ravaioli (2008)	33%	3-y 72%	3-y RFS 82%	3-y 56.6%
Lei (2013)	/	5-y 70.1%	5-y RFS 66.1%	/
Yao (2015)	38.1%	5-y 77.8%	5-y RFS 90.8%	5-y 56.1%
Metha (2018)	36.4%	5-y 79.7%	5-y RFS 87.3%.	5-y 55.4%
Sinha (2019)	35.3% UCSF-DS vs. 83.8% AC-DS	5-y 78.5% UCSF-DS vs. 50% AC-DS	5-y RFS 86.1% UCSF-DS vs. 40% AC-DS	5-y 56.0% UCSF-DS vs. 21.1% AC-DS
Metha (2020)	/	3-y 79.1% UNOS-DS vs. 71.4% AC-DS	3-y RR 12.8% UNOS-DS vs. 16.7% AC-DS	/
Lewandowski (2009)	TACE 31% TARE 58%	3-y 19% TACE vs. 59% TARE	1-y RFS 73% TACE vs. 89% TARE	/
Lee (2020)	67.6%	5-y 83.3%	5-y RFS 83.5%	/
*Studies with no upper limits of downstaging by size and number of tumours but no vascular invasion or extrahepatic spread*				
Mazzaferro (2020)	73%	5-y 77.5% (LT) vs. 31.2% (control)	5-y RFS 76.8% (LT) vs. 18.3% (control)	/
Cillo (2007)	/	3-y 84%	/	3-y 85%
Graziadei (2003)	73.3%	4-y 41%	4-y RR 30%	5-y 31%
Otto (2006)	54.8%	5-y 80.9%	5-y RFS 69.3%	/
De Luna (2009)	63%	3-y 84%	/	3-y 37%
Jang (2010)	41.5%	5-y 54.6%	5-y RFS 66.3%	5-y 10%
Barakat (2010)	56%	/	/	2-y 78%
Green (2013)	77%	1-y 100%	2-y RR 28.5%	/
Toso (2015)	/	4-y 76.6% (to Milan) vs. 100% (to TTV/AFP)	4-y RR 7.4% (to Milan)	4-y 53.8% (to TTV/AFP)
Hangzhou (2020) *	39.5%	3-y 42.2% (group A) 70.7% (group B) 73.5% (group C) 26.5% (group D)	3-y RR 52.3% (group A) 10.3% (group B) 11.6% (group C) 59.4% (group D)	/
Kardashian (2020)	/	5-y 64.3%	5y RFS was 59.5%, and RR was 18.7%	/
*Studies with no upper limits of downstaging for vascular invasion*				
Chapman (2008)	22.3%	5-y 93.8%	5-y RFS 93.8%	/
Chapman (2017)	42%	5-y 85.8% (within UCSF) vs. 66.2% (beyond UCSF)	5-y RFS 87.2% (within UCSF) vs. 62.8% (beyond UCSF)	/
Assalino (2020)	/	5-y 59.6% (83.3% if AFP pre LT < 10 ng/mL)	5-y RFS 56.3% (71.8% if AFP pre LT < 10 ng/mL)	/

AC= All-Comers, AFP = Alpha-fetoprotein, DS = Downstaging, LT = Liver Transplant, RFS = Recurrence Free Survival, RR = Recurrence Rate, TACE = Transarterial chemoembolization, TARE = Transarterial Radioembolization, TTV = Total Tumour Volume, UCSF = University of California San Francisco, San Francisco, CA, USA, UNOS = United Network for Organ Sharing, y = years. * Group A (*n* = 46) Beyond Hangzhou criteria, Hangzhou, China, and downstaging failure; Group B (*n* = 30) Beyond Hangzhou criteria and successful downstaging; Group C (*n* = 113) Within Hangzhou criteria and remain within Hangzhou criteria; Group D (*n* = 17) Within Hangzhou criteria and progress beyond Hangzhou criteria. Hangzhou Criteria (HC): (a) reduction in total tumour diameter ≤ 8 cm or (b) total tumour diameter > 8 cm with grade I or II tumour differentiation but AFP level ≤ 400 ng/mL.

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
