# Peer review of "Upper Limits of Downstaging for Hepatocellular Carcinoma in Liver Transplantation"

_cancers, 2021, doi:10.3390/cancers13246337_

Round 1

Reviewer 1 Report

The Paper is well written and resume all the step that we have followed to obtain a good knowledge in this tumor and treatment. In my opinion this is a review of the litterarture. There are small mistake of formatting in the text

Reviewer 2 Report

There is another national pre-transplant HCC criteria, that has been applied in Japan, which is called “5-5-500”. It should be included.

“Expanded living-donor liver transplantation criteria for patients with hepatocellular carcinoma based on the Japanese nationwide survey: the 5-5-500 rule - a retrospective study.
Transpl Int. 2019 Apr;32(4):356-368. “

Combination of systemic treatment with loco-regional therapy would be one of the major strategies for HCC. Therefore, it should be mentioned in the end of the discussion.

Reviewer 3 Report

The review from Biolato and coll. entitled "Upper limits of downstaging for hepatocarcinoma in liver transplantation" provides a detailed summary of current evidence concerning selection of HCC transplant candidates after tumor downstaging

The work is generally well organised and well written, although there are some recent papers that have been skipped and should be mentioned

1) The Authors should include a recently published paper focusing on application of LI-RADS classification during preoperative workup of LT candidate: 10.1111/tri.13983

2) The recent paper from the Toronto group focusing on machine-learning algorithm for post-transplant HCC recurrence also deserves a mention: 10.1002/lt.26332.

3) The role of des-gamma-carboxy prothrombin/PIVKA as a biological marker of HCC aggressiveness, especially after neoadjuvant treatments, should also be reported: 10.5301/ijbm.5000276; 

4) In the section 4.2, page 6, there is a 561%, that should be corrected (I am sorry but the manuscript is without line numbers)

5) I also noticed this typo at page 7: (Prentice.2016) - is this a missed reference?

6) I would also like some comments concerning the prioritization of LT candidates according to tumor biology, as recently speculated by Mazzaferro (10.1002/hep.28420) and further explored by Di Sandro et al. (10.3390/cancers11060741)

7) Please modify the references style according to journal guidelines

Best regards

Reviewer 4 Report

The present manuscript by Marco Biolato et al. is a comprehensive review about the current limits of downstaging to rescue patients with HCC beyond Milan criteria for LT. The authors have performed an outstanding work to gather the most relevant published data on this issue. However, an additional effort is required to summarize the evidence in order to make the manuscript more attractive.

The authors are kindly invited to consider the following comments:

- The manuscript is probably too long. Introductory statements about selection criteria for liver transplantation could be shortened.

- Tables legends are not available. Tables are difficult to read due to the reduced size of the font.

- I failed to identify table references within the main text.

- The authors could consider transforming table 1 into a figure showing the different expansion criteria in a metroticket format.

- Section 4.2 could be also shortened if the reader is referred to the specific table.

- When defining macrovascular invasion, the authors should clarify that radiological assessment is sometimes misleading and it is difficult, even for experienced radiologists, to differentiate malignant vs non-malignant thrombosis. Some of the successful downstaging reports in patients with previously diagnosed macrovascular invasion could be explained by an inaccurate initial diagnosis. Please comment.

- The conclusion paragraph is too sparse. Please, make our best to conclude with one or two short paragraphs including clear messages.

Round 2

Reviewer 3 Report

The Authors properly addressed my previous comments

Paper quality has been improved in the revised version, that is suitable for publication

Best regards

Reviewer 4 Report

The authors have significantly improved the quality of the manuscript as per previous comments. I congratulate them for the newly added figure, which is very informative.